# TOPIC AND HYPERBOLIC TRANSFORMER TO HANDLE MULTI-MODAL DEPENDENCIES

## ABSTRACT

As multi-modal search relies on jointly learning image-text representations and has been investigated in the literature, our innovation is to develop Chimera, a framework in which to learn their representations and similarities. Because the core of multi-modal search is learning the modalities in a shared semantic space and measuring their similarities, search quality depends on which expressive space is utilized in learning. This motivates us to identify the space that can elucidate their semantic and complex relationships with small information loss. Novelty is assured by introducing the topic and hyperbolic as spaces, and performing contrastive/metric learning tasks to ensure the cooperation of these spaces with Transformer. Experiments show that Chimera empowers pre-trained models for multi-modal search tasks and demonstrate the ability of the layers it introduces.

## 1 INTRODUCTION

While conventional search systems find relevant documents to a query, multi (cross) -modal search systems retrieve relevant instances to answer the query. The text-based image retrieval models return images matched to a given text, where images are ranked based on their similarity to the text. Inferring the semantic alignment between images and words in the associated sentences allows the fine-grained interplay between vision and language data to be captured, and makes image-text matching more interpretable. As this search model needs semantic-intermediate expressions for both images and text, our challenge is to identify spaces that can learn their appropriate representations and define similarities.

Since joint image-text embedding is the bedrock for most Vision-and-Language (V+L) tasks, many studies have explored how to learn their embeddings, project them into a semantic space, and measure their similarity. Many existing multi-modal approaches employ either embedding Lee et al. (2018) or classification Huang et al. (2017). As evidenced by the embedding approach, the semantic space is generally of low dimension allowing representation by vectors and the direct computation of modalities by conventional distance metrics (e.g., cosine similarity or Euclidean distance) as the similarity function in this embedding space. The literature describes the use of a variety of matching functions, such as metric learning Hsieh et al. (2017); Tay et al. (2018b) and/or neural networks He et al. (2017). Although the studies are primarily focused on designing distance functions over objects (i.e., between users and items), and have empirically demonstrated reasonable success in collaborative ranking with implicit feedback, matching functions work only in the scope of Euclidean space Tran et al. (2020). Since there is a hypernym, and an entailment relationship within words, and texts Vendrov et al. (2016), one is able to perceive a semantic hierarchical relationship between sentences and images.

This intuition motivates us to explore spaces that explicitly illuminate both semantic and hierarchical relationships and project them for representation learning. That is, our framework, Chimera, focuses on topics and hyperbolic geometry as the semantic space and the roomier space, respectively; the contrastive and metric learning approach are used as training objectives. As topic models describe the statistical relationships of word occurrences as global information (e.g., over a given data), Chimera adopts their concept to complement Transformer with global information, and mitigate the strong conditional dependencies be-

tween inputs and modalities. As recent studies Hui & Ku (2022); Yang et al. (2022) show that hyperbolic space is roomier than Euclidean space, and can better represent complex relationships, Chimera employs both this space and its operation to realize simpler representations rather than looking for good image/text matches in Euclidean space. Unlike other models, Chimera is a framework in which Transformer-based models are directed to jointly learn representations in different rooms (i.e., topics and hyperbolic space). Chimera applies metric learning in both Euclidean and hyperbolic space to utilize the benefits of both these spaces, and provides higher similarity scores to the positive pairs than the negative ones.

The key advantages of our approach are two-fold:
**Theoretical contribution**: Chimera (1) derives a way to incorporate both topic and hyperbolic space with Euclidean space to learn more semantic and complex relationships over modalities, in an end-to-end fashion. (2) trains objectives in the multi-task learning, and adopts the simplicity and effectiveness of the metric learning paradigm to combine Euclidean space based pre-trained models and hyperbolic space model in the fine-tuning stage.
**Practical contributions**: Chimera is a plug and play framework, that can utilize the benefits of pre-trained models by fusing Euclidean/hyperbolic/topic spaces through a contrastive/metric learning paradigm in fine-tuning, as shown in **4.3** and **4.4**.

## 2 RELATED WORK

Cross-modal or multi-modal retrieval Balaneshin-kordan & Kotov (2018); Carvalho et al. (2018); Wang et al. (2019) finds a set of objects in response to textual queries, or assigns understandable terms to objects; multimodality inputs are processed simultaneously for joint vision + language (V+L) understanding. Recent pre-trained language models Devlin et al. (2019); Radford et al. (2019); Yang et al. (2019); Liu et al. (2019b); Lan et al. (2020), use Transformer Vaswani et al. (2017) for learning contextualized text representations; they have yielded great advances in NLP tasks, and have been applied to multi-modal tasks Singh et al. (2022); Hu & Singh (2021). ViLBERT Lu et al. (2019) and LXMERT Tan & Bansal (2019) introduced the two-stream architecture, where separate Transformers are applied to images and text independently; the results of which are fused by a third Transformer in the second stage. Unicoder-VL Li et al. (2020a) takes the visual regions of the image and textual tokens of the sentence as the input and then encodes the input to yield the linguistic embedding and image embedding, respectively. As with pre-training, VL-BERT Su et al. (2020) performs pre-training on both visual-linguistic and text-only datasets jointly, but does not address the task of Sentence-Image Relationship Prediction unlike other works Lu et al. (2019). UNITER Chen et al. (2020b) uses conditional masking on Masked Language Modeling and Masked Region Modeling, and introduces a novel Word-region Alignment pre-training task. It can realize heterogeneous downstream V+L tasks with joint multi-modal embeddings. Zhou et al. Zhou et al. (2020) present a unified Vision-Language Pre-training model that also allows generation via a unified Transformer with various self-attention masks Dong et al. (2019). Murahari et al. Murahari et al. (2020) proposes VisDial-BERT that employs pre-trained models for visual dialog. VD-BERT Murahari et al. (2020) relies on the self-attention mechanism within a single-stream Transformer encoder to capture the complex interactions in a unified manner. Oscar Li et al. (2020b) is motivated by the observation that the salient objects in an image can be detected, and are often mentioned in the accompanying text, and uses object tags as anchor points to align the image and language modalities in a shared semantic space. VinVL Zhang et al. (2021) feeds the visual features generated by the new object detection model, and utilizes an improved version of OSCAR+ to pre-train the V+L model and fine-tune it for a wide range of downstream VL tasks.

Hyperbolic representation learning has recently demonstrated great potential across a diverse range of applications such as learning entity hierarchies Nickel & Kiela (2017) and/or natural language processing Tay et al. (2018a), hyperbolic graph neural network Chami et al. (2019); Liu et al. (2019a). Hyperbolic geometry has already been applied to the recommender systems domainTran et al. (2020); Wang et al. (2022a) or image recognition applications Liu et al. (2020); Khrulkov et al. (2020). Hyperbolic space is a kind of manifold space studied in Riemannian geometry, in which basic mathematical operations (e.g., distance measurement) are defined differently from Euclidean space. Nickel and Kiela Nickel &

Kiela (2018) propose a method that exploits the individual strengths of both these models by building embeddings using the Lorentz model and mapping them into the Poincaré ball. Hyperbolic Visual Embedding Learning Networks Chen et al. (2020a) learn the hierarchy-aware image embedding features in hyperbolic space, where image labels are projected into hyperbolic space with the Poincaré hierarchy embedding model Nickel & Kiela (2018) and Poincaré Globe Tifrea et al. (2019). Law et.al Law et al. (2019) explain why the squared Lorentzian distance is a better choice than the Poincaré metric.

Despite the advances made in hyperbolic embeddings, how to use hyperbolic embeddings for downstream tasks such as classification is a challenge due to the absence of corresponding hyperbolic neural network layers Hui & Ku (2022). This motivates us to explore a space that identifies more semantic and complex relationships with small information loss, and introduce the topic and hyperbolic as alternative spaces. As with training tasks, Chimera can jointly fine-tune models with other training frameworks Radford et al. (2021); Wang et al. (2022b); Huang et al. (2022); Zeng et al. (2022); Zhong et al. (2022).

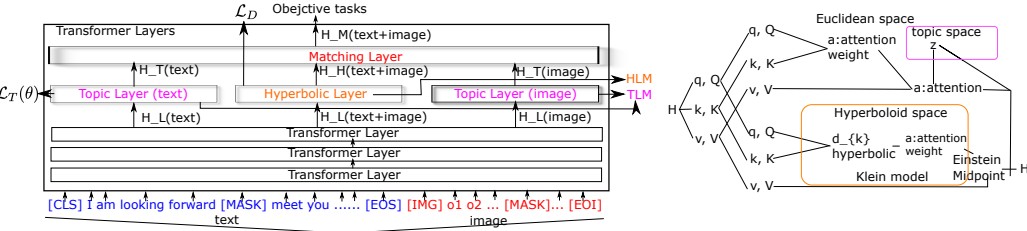

Figure 1: Architecture of (left) Chimera, and (right) various spaces and models adopted in Chimera for the attention layer: Chimera consists of a text/image topic layer, a hyperbolic layer, a matching-layer, and objective function (e.g., $\mathcal{L}_T(\theta)$, $\mathcal{L}_D$, TLM, and HLM) which can be injected into pre-trained Transformer-based models without modifying their architecture. Upon receiving the output of top Transformer layer, $\mathbf{H}_L$, the hyperbolic layer feeds $\mathbf{H}_h$ to the matching layer. The text/image topic layer receives $\mathbf{H}_L(text)/\mathbf{H}_L(image)$ and feeds $\mathbf{H}_T(text)/\mathbf{H}_T(image)$ to the matching layer, where $\mathbf{H}_L(text)$, and $\mathbf{H}_L(image)$ denotes the $\mathbf{H}_L$ associated with text, and image, respectively. That is, Chimera decomposes $\mathbf{H}_L \in \mathbb{R}^{|x| \times d_h}$ into $\mathbf{H}_h$, $\mathbf{H}_T(text)$, and $\mathbf{H}_T(image)$, and integrates them into $\mathbf{H}_M \in \mathbb{R}^{|x| \times d_h}$.

## 3    CHIMERA

### 3.1   MODEL OVERVIEW

The architecture of Chimera is illustrated in Figure 1. Chimera exists between the top of Transformer layers and model specific objectives or output functions. It receives the hidden states of last Transformer layer as its input, transforms them via the topic and Hyperbolic space with objectives, and feeds its output into the model-specific objectives or functions.

As Chimera is based on the understanding that hyperbolic space can better represent complex relationships than Euclidean space Nickel & Kiela (2018); Zhang & Gao (2020); Gülçehre et al. (2019), it represents sets of images and texts in both a topic and a hyperbolic space, to illuminate their semantic and hierarchical relationships. This direction leads us to adapt Transformer-based models for performing an attentive read operation over different representation vectors in both topic and hyperbolic space, while maintaining the simplicity and effectiveness of the metric learning paradigm. In practice, this framework enables Chimera to connect topics, and Euclidean and hyperbolic space, while retaining the benefits of pre-trained models and alleviating catastrophic forgetting Ramasesh et al. (2021).

### 3.2   TRANSFORMER LAYERS

Transformer has $l$ layers, each of which contains two blocks. While the core of the first block is multi-head attention with $k$-heads, the core of the second block is a feedforward network with ReLU activation Nair & Hinton (2010); it embeds inputs with $|x|$ length, $\mathbf{X} \in \mathbb{R}^{|x|}$, to the hidden representation, $\mathbf{H}_0 \in \mathbb{R}^{|x| \times d_h}$, and projects it using inner dimension $f$, with layer normalization Ba et al. (2016):

$$\mathbf{H}_0 = EMB(\mathbf{X}), \quad \bar{\mathbf{H}}_{l-1} = LayerNorm(\mathbf{H}_{l-1}), \quad \mathbf{H}_l = MultiHead(\bar{\mathbf{H}}_{l-1}) + \bar{\mathbf{H}}_{l-1},$$
$$MultiHead(Q, K, V) = [h_1; \cdots; h_k]\mathbf{W}_o, \quad \bar{\mathbf{H}}_l = LayerNorm(\mathbf{H}_l), \tag{1}$$
$$\mathbf{H}_{l+1} = FFN(\bar{\mathbf{H}}_l) + \bar{\mathbf{H}}_l, \quad FFN(\bar{\mathbf{H}}_l) = max(0, \bar{\mathbf{H}}_l U)V, \quad h_j = Attention(Q_j, K_j, V_j),$$

where $\mathbf{U} \in \mathbb{R}^{d_h \times f}$, $\mathbf{V} \in \mathbb{R}^{f \times d_h}$, and $\mathbf{W}_o \in \mathbb{R}^{kd_h \times d_k}$ are learnable weights, $\mathbf{Q}_j, \mathbf{K}_j, \mathbf{V}_j \in \mathbb{R}^{|x| \times d_k}$ are obtained by transforming the output of the $l$-th layer, $\mathbf{H}_l$, using $\mathbf{W}_l^{Qj}, \mathbf{W}_l^{Kj}, \mathbf{W}_l^{Vj} \in \mathbb{R}^{d_h \times d_k}$, respectively.

## 3.3 TOPIC LAYER

The motivation for introducing the topic layer is to model the uncertainty in the generative process, and strike a balance between visual information in the image and linguistic knowledge acquired from the text. Given input sequence $\mathbf{x}_d = \{x_{d,1}, \cdots, x_{d,|x|}\}$ and dataset $D = \{\mathbf{x}_1, \cdots, \mathbf{x}_D\}$, non-autoregressive generation can be achieved by minimizing the following independence assumption in the decoding process:

$$min_\theta \mathcal{L}(\theta) = min_\theta - \sum_{d=1}^{|D|} \sum_{t=1}^{|x|} \log P_\theta(x_{d,t}|\mathbf{x}_{d,\backslash t}), \tag{2}$$

where $\theta$, and $\mathbf{x}_{d,\backslash t}$ represents model parameters, and $\mathbf{x}_d$ without $x_{d,t}$, respectively.

To mitigate the gap between inputs and modalities, our framework introduces topics, $z$, into this process and so modifies Eq (2) to:

$$min_\theta \mathcal{L}_T(\theta) = min_\theta - \sum_{d=1}^{|D|} \sum_{t=1}^{|x|} \log \sum_{z_t=1}^{Z} P_\theta(x_{d,t}|\mathbf{x}_{d,\backslash t}, z_t) P_\theta(z_t|\mathbf{x}_{d,\backslash t}), \tag{3}$$

where $z_t$ denotes the topic of the $t$-th token, and $Z$ is the number of topics. That is, $P_\theta(z_t|\mathbf{x}_{d,\backslash t})$ is the prior distribution over latent topic $z$, and $P_\theta(x_{d,t}|\mathbf{x}_{d,\backslash t}, z_t)$ is the "generative" distribution over words, $\mathbf{V}$. This formulation ensures that sequence $\mathbf{x}$ can be generated by a random process involving latent topic variable $z$: (1) $z_t$ is first generated from the conditional distribution $P_\theta(z_t|\mathbf{x}_{d,\backslash t})$. (2) $x_{d,t}$ is finally generated from $P_\theta(x_{d,t}|\mathbf{x}_{d,\backslash t}, z_t)$. Note that $P_\theta(z_t|\mathbf{x}_{d,\backslash t})$ is a multinomial distribution over discrete variables, not The Gaussian distribution used in variational autoencoders Kingma & Welling (2014).

By placing this topic layer on both the text and the visual encoder (Transformer layers), we mitigate the discrepancy between the cross-modal representations learned by the encoder and the representation needed by the decoder for generating text. As Chimera injects topics on the top blocked attention layers, it maps hidden representation vector $\mathbf{h} \in \mathbb{R}^{d_h}$ to latent topic vector $\mathbf{z} \in \mathbb{R}^Z$, and then projects this topic vector into the topic-specific distribution over words. This yields Eq (1) by defining topic matrix, $\mathbf{W}_Z \in \mathbb{R}^{d_h \times Z}$, and word generation function, $\mathcal{F}(\mathbf{h}_{L,t})$, where $V$ is the size of the vocabulary. This method is used to sample the next token according to the following probability:

$$\mathbf{H}_t(text/image) = \underbrace{LayerNorm(\mathbf{H}_L)\mathbf{W}_Z}_{P_\theta(z_t|\mathbf{x}_{d,1:t-1})} \times \underbrace{\mathcal{F}(\mathbf{h}_{L,t})|z=z_t)}_{P_\theta(x_{d,t}|\mathbf{x}_{d,1:t-1}, z_t)}, \tag{4}$$

where $\mathbf{W}_Z$ is a learnable weight. As with $F_z \in \mathcal{F}(\mathbf{h}_{L,t})|z=z_t)$, we propose three transformations to generate $x_{d,t}$ that accords with the given $z_t$ and $\mathbf{x}_{d,1:t-1}$.

$$F_z = \mathbf{W}_{V|Z} \times \begin{cases} \mathbf{h}_{L,t} & \text{residual if } z = 0 \\ (1-\omega)\mathbf{h}_{L,t} + \omega g_z & \text{addition if } z > 0 \\ \mathbf{h}_{L,t} \otimes g_z & \text{multiplication if } z > 0 \\ \mathbf{h}_{L,t}\mathbf{W}_{R|z} + \mathbf{b_z} & \text{affine if } z > 0, \end{cases} \tag{5}$$

where $g_z \in \mathbb{R}^{d_h}$, $\mathbf{W}_{R|z} \in \mathbb{R}^{d_h \times d_h}$, and $\mathbf{b}_z \in \mathbb{R}^{d_h}$ are the learnable weights specific to topic $z$. We prepare the residual to select the input if $z = 0$, to select Euclidean space. Note that just as Eq (2) is transformed into Eq (3) through the introduction of topics, $\mathbf{W}_{V|Z} \in \mathbb{R}^{d_h \times V}$ used in previous Transformer is decomposed into the product of $\mathbf{W}_Z$ and $\mathcal{F}(\mathbf{h}_{L,t})$ in Eq (4).

### 3.4 HYPERBOLIC LAYER: LORENTZ MODEL AND ATTENTION MECHANISM

Formally, $n$-dimensional hyperbolic space is a manifold of constant negative curvature. As the number of objects grows exponentially with semantic distance from the query, hyperbolic geometry can, unlike Euclidean geometry, encode those objects without creating interference Gülçehre et al. (2019). Among the five most common models for hyperbolic space, we primarily make use of the Lorentz model because its distance function avoids numerical instabilities, unlike the Poincaré equivalent. Note that one-to-one isometric transformations can be defined between each different model of hyperbolic space. The Lorentz model is the only unbounded hyperbolic model and is defined as $\mathcal{L}^n = (\mathbb{H}^n, g_{\mathcal{L}})$ with points constrained by $\mathbb{H}^n = \{x \in \mathbf{R}^{n+1} : \langle \mathbf{x}, \mathbf{x} \rangle_{\mathcal{L}} = -1, x_0 > 0\}$, where the Riemannian metric tensor is $g_{\mathcal{L}}(x) = diag(-1, 1, ..., 1)$, and $\langle \mathbf{x}, \mathbf{y} \rangle_{\mathcal{L}}$ is the Lorentzian scalar product. The associated distance function on $d_{\mathcal{L}}$ is then given as:

$$d_{\mathcal{L}}(\mathbf{x}, \mathbf{y}) = arcosh(-\langle \mathbf{x}, \mathbf{y} \rangle_{\mathcal{L}}), \quad \langle \mathbf{x}, \mathbf{y} \rangle_{\mathcal{L}} = \Sigma_{i=1}^n x_i y_i - x_0 y_0 \tag{6}$$

As shown in Eq (1), $h_j$ corresponds to scaled dot-product attention, and is written as $\mathbf{R} = softmax(\frac{\mathbf{Q}\mathbf{K}^T}{\sqrt{d}})\mathbf{V}$, where $\mathbf{Q}, \mathbf{K}, \mathbf{V}$ denotes the *queries*, *keys*, and *values*, respectively; $d_h$ is the shared dimensionality of the queries and keys. The attentive read operation over keys, $\mathbf{k}$, by query $\mathbf{q}$ has the following form:

$$r_i = \Sigma_j [\frac{\alpha_{i,j}}{Z}] v_{ij}, \quad \alpha_{i,j} = \exp(\frac{1}{\sqrt{d}} \langle \mathbf{q_i}, \mathbf{k_j} \rangle), \quad v_{ij} = v_j, \quad Z = \Sigma_j \alpha_{i,j}, \tag{7}$$

where $\mathbf{q}_i$ is a vector called the query and $\mathbf{k}'_j s$ are the keys for the memory locations being read from, $\alpha_{i,j}$ is the function that computes a scalar matching score between $\mathbf{q}_i$ and $\mathbf{k}_j$, $v_{ij}$ is a value to be read from $j$-th location by $i$-th query, and $Z(> 0)$ is a normalization factor for the full sum.

As the visual/linguistic features are extracted as vectors in the Euclidean space, they are projected into the hyperbolic space to align with the word representation in this phase. The most natural way to exploit hyperbolic geometry for matching pairs of points is to use the hyperbolic distance between them Gülçehre et al. (2019). To gain the weighted value in the Euclidean Transformer, we project $v_{i,j}$ on Klein model and adopt the Einstein midpoint to compute the aggregation weights appropriately Ungar (2005) and Eq (6). Thus, the hyperbolic attention weights, $\alpha_{i,j}$, and each entry, $r_{i,\mathcal{L}}$, of $\mathbf{R}$ that result from an attentive read can be expressed as:

$$r_{i,\mathcal{L}} = \Sigma_j [\frac{\alpha_{i,j}\gamma(v_{i,j})}{\Sigma_l \alpha_{i,l}\gamma(v_{i,l})}] v_{ij}, \quad \alpha_{i,j} = f(-\beta d_{\mathcal{L}}(\mathbf{q_i}, \mathbf{k_j}) - c), \quad \gamma(v_{i,j}) = \frac{1}{\sqrt{1 - \|v_{i,j}\|^2}} \tag{8}$$

where $\gamma(\cdot)$ are the Lorentz factors, and $f(\cdot)$ denotes the sigmoid function; $\beta$ and $c$ are manually set parameters and learned along with the rest of the network. The above yields $\mathbf{H}_h$.

### 3.5 MATCHING LAYER

This layer is to integrate the output of the topic layers and the Hyperbolic layer into the hidden representation, $\mathbf{H}_M$, which can be fed to objective tasks of previous Transformer based models. Thus, the output of this layer, $\mathbf{H}_l$, can be expressed as:

$$\mathbf{H}_M = (1 - \omega_t)\mathbf{H}_h + \omega_t(\omega_v \mathbf{H}_T(image) + (1 - \omega_v)\mathbf{H}_T(text)) \tag{9}$$

where $\omega_t$ and $\omega_v$ is the learnable weights of topic and visual, respectively. That is, both the topic layers and the hyperbolic layer can be adopted by and work seamlessly within existing Transformer based models.

### 3.6 TRAINING TASKS

**Topic Level Matching (TLM)**: Inspired by contrastive learning Khosla et al. (2020); Radford et al. (2021), Topic Level Matching (TLM) aims to bring similar instances closer

and push away dissimilar instances further from each other. Assuming that similar instances in the Euclid space are considered similar at the topic level, topics are learned to explain the latent semantic similarity; accordingly, we apply TLM to the topic layer. We use $t$, and $i$ to refer to text and image on the topic layer, respectively.

As each image has multiple sentences and these sentences refer to the same image, they are considered to be semantically similar. Since this task aims to utilize this semantic similarity for representation learning, it is designed to predict whether one linguistic sentence can be semantically matched with other sentences attached to the same image. Like Image-Text Matching (ITM) in other models Chen et al. (2020b), this model samples both sentences (positive) with the same image and those with the other images (negative) and learns their matching scores, where it creates the negative pair by replacing the sentence in a paired sample with a randomly-selected equivalent from the other sample.

$$\mathcal{L}_{TLM}(\zeta) = -E_{(t,i)\sim D} \log \frac{\exp \phi_\theta(t,i)}{\exp \phi_\theta(t,i) + \sum_{n=1}^{N} \exp \phi_\theta(t,i_n)}, \tag{10}$$

where $\phi_\theta(t,i)$ denotes the inner product of $t$ and $i$, $N$ is the size of negative samples, and $\{i_n\}_{n=1}^{N}$ are $N$ negative samples.

**Hyperbolic Level Matching (HLM)**: To realize fine-tuning for image-text and text-image retrieval, we formulate the task as a ranking problem. The fine-tuning inputs share the same data preprocessing procedures used in pre-training, except that we do not mask words or regions in the fine-tuning stage. Following the V+L task, we denote the score function as $s$. Here, we are motivated by Visual Linguistic Matching (VLM) in Li et al. (2020a), and ITM. Both define the similarity between two inputs with the ranking loss constraint widely used in embedding-based methods to construct a bi-directional max-margin ranking loss instead if treating the "match" and "mismatch" as a binary classification problem. Similar to Hsieh et al. (2017), for a given query, the nearest neighborhood images are: 1) the images previously referred to by this query, and 2) the images referred to by other words that have similar semantics to this word. That is, we are also able to indirectly observe the relationships between word-word pairs and image-image pairs through the pull-push mechanism of metric learning in Euclidean space.

Using the exponential map allows us to project image features $v_i$ into the hyperbolic space. Here, we transform this feature vector into its polarity form as $(\mathbf{d}, r) \in \mathbb{R}^{n+1}$, where $r = ||v_i||$ and $\mathbf{d} = \frac{v_i}{r}$, i.e. $||\mathbf{d}|| = 1$, and denote this transformed feature as $\tilde{v}_j$. Similarly, we get the transformed feature, $\tilde{w}_i$, from the word embedding $w_i$. Thus, we can project these different features in the same Lorentz space. As the model is trained in Lorentz hyperbolic space, we use the distance, $d_{\mathcal{L}}$, defined in Eq (6) and define our pull-and-push loss function as:

$$\mathcal{L}_{HLM} = \sum_{(i,j)\in D} \sum_{(k,j)\notin D} [m + d_{\mathcal{L}}^2(\tilde{w}_i, \tilde{v}_j) - d_{\mathcal{L}}^2(\tilde{w}_k, \tilde{v}_j)]_+, \tag{11}$$

where $D$ contains all observed implicit feedback, i.e. positive image-word pairs; $[z]_+ = max(0, z)$ is the standard hinge loss and $m > 0$ is the safety margin size.

Because the goal when embedding one space into another is to preserve distances while maintaining complex structures/relationships Sala et al. (2018), HyperML Tran et al. (2020) explores metric learning in hyperbolic space for recommender systems. It defines the distortion optimization function while embedding user-item pairs into hyperbolic space with the constraint of preserving good structure quality for metric learning. As our goal is to coordinate the Euclid space with the Hyperbolic space, our approach adapts this distortion optimization function as follows:

$$\mathcal{L}_{\mathcal{D}} = \sum_{(i,j)\in D} [\frac{|d_{\mathcal{L}}(\tilde{w}_i, \tilde{v}_j) - d_{\mathbb{E}}(w_i, \hat{v}_j)|}{d_{\mathbb{E}}(w_i, \hat{v}_j)}]_+ + \sum_{(k,j)\notin D} [\frac{|d_{\mathcal{L}}(\tilde{w}_k, \tilde{v}_j) - d_{\mathbb{E}}(w_k, \hat{v}_j)|}{d_{\mathbb{E}}(w_k, \hat{v}_j)}]_+, \tag{12}$$

where $|\bullet|$ defines the absolute value, and $\hat{v}_j$ is the embedding representation in Euclid space. Thus, fine-tuning aims to preserve the distances by minimizing $\mathcal{L}_{\mathcal{D}}$, as this function associates lower distortion with better preservation.

Table 1: Basic statistics of the datasets used in fine-tuning

| Datasets | #Training | #Validation | #Test |
|---|---|---|---|
| Flickr30 | 30K | 1K | 1K |
| MSCOCO | 83K | 5K | 5K |

## 3.7 Fine-Tuning

Chimera can adopt pre-trained Transformer-based models and can be fine-tuned for their downstream multi-modal tasks, because Topic/Hyperbolic/Matching Layer and TLM/HLM/$\mathcal{L}_\mathcal{D}$ can be easily plugged into these models without changing their structure. By performing the pre-training over a suitable visual-linguistic corpus, Chimera can be customized and applied to visual-linguistic tasks, while keeping the size of the input and output representations in line with those of the pre-trained Transformer-based models.

While jointly training tasks, $\mathcal{L}_T(\theta)$, $\mathcal{L}_{TLM}$, $\mathcal{L}_{HLM}$ and $\mathcal{L}_\mathcal{D}$ contribute to boost the model performance while providing good representations; there is, however, an unavoidable trade-off between these functions Sala et al. (2018); Tran et al. (2020). Accordingly, we integrate these tasks, the model-specific objective task set, $\mathcal{L}_\mathcal{O}$, in an end-to-end fashion in a multi-task learning framework. The objective function is defined as:

$$\min_\Theta \mathcal{L}_\Theta = \underbrace{\lambda_T \mathcal{L}_T(\theta) + \lambda_{TLM}\mathcal{L}_{TLM} + \lambda_{HLM}\mathcal{L}_{HLM} + \lambda_D\mathcal{L}_\mathcal{D}}_{Chimera} + \underbrace{\lambda_O\mathcal{L}_\mathcal{O}}_{pre-trained models}, \quad (13)$$

where $\Theta$ is the total parameter space that covers all embeddings and variables of the networks, and $\lambda_*$ is the weight of each task.

# 4 EXPERIMENTS

## 4.1 Datasets

We use two datasets, MSCOCO Lin et al. (2014) and Flickr30 [1], a widely used dataset for multi-modal search Wang et al. (2019); Li et al. (2020a), as follows. We follow Karpathy & Li (2015),Faghri et al. (2018) to split these datasets into training/validation/test subsets. Their statistics are shown in Table 1.

## 4.2 Implementation Detail

We implemented Chimera using Pytorch [2] with Nvidia Apex [3], and Horovod [4] to speed up training. We will release this code later. As shown in Figure 1, Chimera can be inserted as additional layers with original objective tasks, $\mathcal{L}_\mathcal{O}$ in Eq (13), into various pre-trained Transformer models without changing their architecture. Chimera works with the Transformer layers and their objective tasks. To get the best out of the combination of the pre-trained models and Chimera rather than comparing fairly over various models, we follow the settings of each pre-trained model.

**Experimental design**: As with Transformer-based models (UNITER, VL-BERT, VD-BERT, ViT)), we modify the implemented models[5] and optimize their parameters. With regard to training Transformer in our framework, we use Adaptive Moment Estimation (Adam) Kingma & Ba (2015) with $\beta1 = 0.9$ and $\beta2 = 0.999$ used for optimization, over mini-batches to update parameters and adopt the dropout strategy Srivastava et al. (2014) to optimize networks. Then we follow the fine-tuning of Radford et al. (2021); Jia et al. (2021) and set number $N$ in Eq (10) to the batch size. After performing hyperparameter

---

[1] `http://bryanplummer.com/Flickr30kEntities/`

[2] `https://pytorch.org/`

[3] `https://github.com/NVIDIA/apex`

[4] `https://github.com/horovod/horovod`

[5] `https://github.com/ChenRocks/UNITER,https://github.com/jackroos/VL-BERT,https://github.com/salesforce/VD-BERT.git,https://github.com/jeonsworld/ViT-pytorch`

Table 2: EX1: The contribution of Chimera in image-text matching tasks over Flickr30 and MSCOCO datasets using $R@N$: The value is the improvement ($+\%$).

| Data | Flickr30 | | | | | | MSCOCO | | | | | |
|---|---|---|---|---|---|---|---|---|---|---|---|---|
| Method | I2T | | | T2I | | | I2T | | | T2I | | |
| ($N =$) | 1 | 5 | 10 | 1 | 5 | 10 | 1 | 5 | 10 | 1 | 5 | 10 |
| UNITER | 1.6 | 3.1 | 8.3 | 2.5 | 4.8 | 10.1 | 1.8 | 3.6 | 8.9 | 2.9 | 5.4 | 11.4 |
| VL-BERT | 1.4 | 3.3 | 8.4 | 2.3 | 4.6 | 9.8 | 1.6 | 3.3 | 9.2 | 2.7 | 5.2 | 11.2 |
| ViLBERT | 1.5 | 3.4 | 8.2 | 2.4 | 4.4 | 9.6 | 1.5 | 3.1 | 9.1 | 2.8 | 5.0 | 11.1 |
| VD-BERT | 1.4 | 3.2 | 8.1 | 2.2 | 4.3 | 9.7 | 1.5 | 3.2 | 9.1 | 2.8 | 5.1 | 10.9 |
| ViT | 1.5 | 3.3 | 8.2 | 2.2 | 4.4 | 9.8 | 1.5 | 3.3 | 9.2 | 2.7 | 5.0 | 11.1 |

Table 3: EX2: The contribution of Chimera in image-text matching tasks over Flickr30 and MSCOCO datasets: Different from the previous experiment shown in Table 2 using $R@N$, we randomly shuffle the order of the words in each input sentence. The value is the improvement ($+\%$)

| Data | Flickr30 | | | | | | MSCOCO | | | | | |
|---|---|---|---|---|---|---|---|---|---|---|---|---|
| Method | I2T | | | T2I | | | I2T | | | T2I | | |
| ($N =$) | 1 | 5 | 10 | 1 | 5 | 10 | 1 | 5 | 10 | 1 | 5 | 10 |
| UNITER | 1.9 | 3.9 | 8.9 | 3.2 | 5.7 | 12.2 | 2.1 | 4.1 | 9.8 | 3.5 | 6.2 | 12.6 |
| VL-BERT | 1.8 | 3.7 | 9.1 | 3.1 | 5.8 | 12.1 | 2.2 | 4.2 | 10.1 | 3.4 | 6.3 | 12.4 |
| ViLBERT | 1.7 | 3.6 | 9.0 | 3.0 | 5.7 | 12.2 | 2.1 | 4.1 | 10.0 | 3.2 | 6.2 | 12.4 |
| VD-BERT | 1.7 | 3.7 | 9.1 | 3.1 | 5.5 | 12.1 | 2.1 | 4.2 | 10.1 | 3.3 | 6.3 | 12.5 |
| ViT | 1.7 | 3.6 | 9.0 | 3.0 | 5.5 | 12.2 | 2.1 | 4.1 | 10.0 | 3.2 | 6.2 | 12.4 |

optimization using grid search, we set them as $\lambda_T, \lambda_{TLM}, \lambda_{HLM}, \lambda_D$, and $\lambda_O$=0.5, 0.5, 0.5, 0.5, and, 1.0, and apply these values to other models. Like other state-of-the-art pre-trained V-L models, our framework can fine-tune for various downstream visual-linguistic tasks, by simply modifying the input format, output prediction, loss function and training strategy.

### 4.3 Evaluation metrics and Results

For the V-L task, we perform two image-text matching tasks:
**I2T**: sentence retrieval, i.e., retrieving ground truth sentences given a query image.
**T2I**: image retrieval, i.e., retrieving ground truth images given a query text. Since these tasks are based on rankings, candidates are ranked according to their scores, and the number of candidates varies with the query. For both tasks, the commonly used evaluation metric is R@N, which is defined as the recall rate for the top $N$ results for the query.

Table 2 shows the results of these tasks, **EX1**. We confirm that the contribution of Chimera is proportional to the increase in $N$; this tendency is stronger for T2I than I2T.

As the accuracy was virtually the same in the previous experiment, we performed an additional experiment with different conditions. That is, we randomly permuted words in each sentence to determine the robustness of the model. As shown in Table 3, Chimera offers significantly better performance than the other models than in experiment **EX2**. This table indicates that not only the hyperbolic space but also TLM/HLM, which are newly introduced as the training tasks in our architecture, contributed to this superiority, as shown by the ablation analysis. Since the difference is greater for T2I than for I2T, the latter task is considered to be a more difficult than the former. This result suggests that hyperbolic space is better suited to the visual-linguistic task. Chimera is more robust against word reordering because Topic and Hyperbolic layers learn representations on each input level.

### 4.4 Ablation analysis

To investigate the respective contributions of pre-training tasks to overall performance, we conducted an ablation analysis over the same datasets with the same metric following EX1 2. In order to compare the effect of each component, Chimera used ViT as the pre-trained model. To emphasize the difference between the effects of these components, we performed fine-tuning over the data with 20% of each of the images and texts randomly masked, the

Table 4: Ablation analysis of pre-training of image-text matching tasks over Flickr30: In this table, TL, HL, Z, and N denote Topic Layer, Hyperbolic Layer, #topics, and # dimensions of Hyperbolic space, respectively. In TLM, ad, mu, and af denote addition, multiplication, and affine in Eq (3), respectively. The best results are marked in bold, where the bold value denotes the statistical significance for $p < 0.01$, compared to the best baseline. TLM and HLM/$\mathcal{L}_\mathcal{D}$ need the topic layer and the Hyperbolic layer, respectively.

| Data | | | | | Flickr30 | | MSCOCO | |
| --- | --- | --- | --- | --- | --- | --- | --- | --- |
| | Tasks | | | | I2T | T2I | I2T | T2I |
| TL | | HL | | $\mathcal{L}_\mathcal{D}$ | R@1,5,10 | R@1,5,10 | R@1,5,10 | R@1,5,10 |
| $Z$ | TLM | $N$ | HLM | | | | | |
| 20 | af | 20 | w | w | 67.8,**91.2**,97.1 | **58.1**,77.1,**89.9** | 66.5,**91.4,97.5** | 58.7,**77.6,90.2** |
| 20 | ad | 20 | w | w/o | 64.8,88.2,92.1 | 56.9,75.3,86.8 | 64.2,84.8,92.2 | 54.8,74.2,86.1 |
| 20 | mu | 20 | w | w/o | 66.1,89.1,96.3 | 57.3,75.9,87.9 | 65.1,86.6,93.5 | 55.6,75.3,87.2 |
| 20 | af | 20 | w | w/o | 64.2,87.6,90.3 | 56.5,74.8,86.2 | 63.7,84.1,91.7 | 53.9,73.1,85.2 |
| 20 | w/o | 20 | w | w | 67.2,90.1,96.4 | 56.9,76.2,88.8 | 66.3,90.3,96.6 | 58.3,76.8,89.4 |
| 10 | af | 20 | w | w | 65.1,88.2,91.2 | 57.1,75.7,86.9 | 64.2,84.8,92.9 | 54.2,73.6,85.7 |
| 0 | w/o | 20 | w | w | 63.2,84.1,88.3 | 56.2,75.2,86.1 | 63.4,83.9,92.1 | 53.7,73.1,85.2 |
| 20 | af | 20 | w/o | w/o | 67.5,90.3,97.0 | 57.2,76.8,88.9 | 66.3,90.8,96.9 | 58.4,76.9,89.5 |
| 20 | af | 10 | w | w | 67.1,90.2,96.8 | 56.8,76.8,88.8 | 65.8,88.4,95.7 | 57.3,76.1,88.4 |
| 20 | af | 0 | w/o | w/o | 66.7,89.8,96.2 | 56.3,76.7,89.3 | 65.2,87.9,94.2 | 56.7,75.7,87.8 |

order of the words in each sentence were randomly shuffled. A comparison of improvements achieved is shown in Table 4. The lower the value, the greater the effectiveness of that excluded task. A comparison of the magnitude of the decrease shows that the contribution of both the Topic layer and the Hyperbolic layer to Chimera is higher, and supports our assumption that both topic space and hyperbolic space contribute to providing better representations through the discovery of the semantic relationships and the summarization of the complex relationships. This table shows that among these tasks, the topic layer is more sensitive to both topic specific tasks (e.g., TLM and HLM) and a decrease in the number of topics, which corresponds to the number of Hyperbolic dimensions, than the Hyperbolic layer.

## 5 DISCUSSION

Our approach uses topic and hyperbolic space and metric learning to focus on the characteristics and differences of multi-modal dependencies. Each image and its corresponding text differ not only in the number of features but also in the amount of information they contain. The relationship between images and text is asymmetric, and this relationship is observed as task asymmetry in Table 3, **EX2**. Table 3 implies that the meaning of a sentence in a text does not change much even if a word is replaced, but images lose their meaning if regions are shuffled in each image, that is the degree of freedom is low. As the V+L task requires a representation space into which both images and text can be embedded without any other bias, our architecture learns their representations and projects them in topic and hyperbolic space. To jointly learn image-text representations more effectively, Chimera employs metric learning to coordinate representations in the Euclid space with those in the hyperbolic space, and can move a data point a certain distance in hyperbolic space with smaller force than is needed in Euclidean space.

## 6 CONCLUSION

To develop semantic intermediate expressions for both images and texts, our proposed framework, Chimera, bridges the gap between Euclidean and hyperbolic geometry through topics and the metric learning approach. This leads our framework to project image-text representations into topic and hyperbolic space while preserving semantic similarity in the Euclidean space through metric learning. Experiments on various data sets showed that Chimera achieved better results on the Vision-and-Language task than the baselines and showed its ability to learn the image-text dependencies.

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
