# OpenReview forum: "Topic and Hyperbolic Transformer to Handle Multi-modal Dependencies"
_ICLR.cc/2023/Conference — Submitted to ICLR 2023_

### Official Review · Reviewer_Gm2Z · 2022-10-21

**Confidence:** 3
**Clarity, Quality, Novelty And Reproducibility:** 1. The structure and writing of this …
**Correctness:** 3
**Technical Novelty And Significance:** 3
**Empirical Novelty And Significance:** 2
**Recommendation:** 5

**Strength And Weaknesses:**

Strength
+ This paper combines the topic model, hyperbolic space and Euclidean space to better model the complex relationships between modalities.
+ The proposed Chimera method is plug-and-play. Diverse experiments are provided, validating that it can benefit the existing pre-trained models.

Weaknesses
- The authors claim that the hyperbolic space can better learn the complex cross-modal association. However, there is lacking strong evidence to support it. In table 4, the results of w HL (hyperbolic space) do not consistently outperform its w/o HL (hyperbolic space) counterpart. In addition, experiments only include the module effectiveness of HL. If using attention module in the Euclidean space replaces HL, could similar results be achieved?
- In the introduction section, it states that Chimera has theoretical contribution to learn more semantic and complex relationships over modalities. However, it seems that the method section only provides the statement of the proposed framework, and the theoretical support does not well proved.
- The ablation studies about the hyper-parameters are missing. Does the selection of hyper-parameters affect the model performance a lot?
- This paper focuses on multi-modal representation learning, but the experiments only contain vision + language dataset.


**Summary Of The Paper:**

In this paper, the authors focus on the multi-modal representation learning and cross-modal similarity estimation topic and propose the Chimera framework. The proposed framework uses the hyperbolic space to better represent complex multi-modal relationships. The topic model is also utilized. The proposed Chimera method is plug and play. Experiments verify its effectiveness when combine with existing multi-modal representation learning framework.

**Summary Of The Review:**

This paper proposes a plug-and-play framework, Chimera, for enhancing the multi-modal representation learning and cross-modal similarity estimation. The proposed method can improve the performance of pre-trained modal. However, the novelty of this paper both technical and empirical is limited. More detail is stated in the last part.

---

> ### Author Response · Authors · 2022-11-12
> **Our current responses**
>
> Thank you for reviewing our manuscript.
> These are our responses at this stage.
> If there is anything missing, we would be very grateful and will answer them if you could let us know.
> We resubmitted the paper with increased resolution by changing the image format to PDF instead of PNG.
>
> W1) The authors claim that the hyperbolic space can better learn the complex cross-modal association. However, there is lacking strong evidence to support it. In table 4, the results of w HL (hyperbolic space) do not consistently outperform its w/o HL (hyperbolic space) counterpart.\
> --->The results with HL outperforms their without HL under other conditions being equal (i.e., $Z$, TLM, N, = 20, af, 20), but please tell us which column you are comparing.
>
> W2) In addition, experiments only include the module effectiveness of HL. If using attention module in the Euclidean space replaces HL, could similar results be achieved?.\
> --->Unless we misunderstand your question, TL is a Euclidean space in this framework and uses the attention module.
> Since $\mathcal{L}_{\bf{\mathcal{D}}}$ is a function of HL, it cannot operate without HL.
>
> W3) In the introduction section, it states that Chimera has theoretical contribution to learn more semantic and complex relationships over modalities. However, it seems that the method section only provides the statement of the proposed framework, and the theoretical support does not well proved.\
> --->The elemental technologies (i.e., Hyperbolic representation and modeling) and their results are described in 2 RELATED WORK.
>
> W4) The ablation studies about the hyper-parameters are missing. Does the selection of hyper-parameters affect the model performance a lot?\
> --->Although this possibility cannot be ruled out,
> we applied Ray Tune into hyper-parameter tuning, and did not consider these tuning and their effect in the ablation studies.
>
> W5) This paper focuses on multi-modal representation learning, but the experiments only contain vision + language dataset.\
> C1) The novelty of this paper is limited. For technical novelty, this proposed framework is combination of existing techniques (topic model, hyperbolic space, transformer, contrastive learning, ranking loss). Also, there is not enough technical support for such combination.\
> ---> There is novelty not only in this combination, but also in the rationale for the combination and the task application of the components.
> In a nutshell, topics are considered as a quantized sample of the underlying feature distribution, and hyperbolic models allow to line up text and visual features in a more compact alternative space.
> Firstly, the motivation is the first approach to seek an alterative space that can elucidate their semantic and complex relationships with small information loss, rather than conventional latent hidden spaces. As explained in 3.3, the topic layer is not just an application of topic models. We derive both several variants sharing its concept, and its training task in 3.6. Then we compare them in the ablation analysis, as shown in Table 4, and determine ''affine'' adapted to our goal. As with the hyperbolic layer, we select Lorentz model from common hyperbolic models and derived this layer according to Transformer and its training task in 3.6.. Finally, we design objective function for these different spaces to learn the space in a cooperative manner, and align both text and image on each space.
>
> C2) For empirical novelty, the experiments are not inadequate and diverse enough. Only two V+L datasets are used. The downstream task only focuses on image-text matching.\
> --->This may appear so because our experimental strategy is not only to follow the design of previous studies, but also to present results narrowly compared to SOTA (i.e., VD-BERT and ViT) in order to improve reproducibility for other researchers and to reduce the burden on reviewers.

---

> > ### Comment · Reviewer_Gm2Z · 2022-11-18
> > **Updated review**
> >
> > Thanks for the authors' efforts in response and some of my questions are solved. But there are still some issues:
> >
> > Q1) The w/ HL improvement at the same condition (Z, TLM, N, = 20, af, 20) are very slight in about half cases (5 column with no more than 0.3, and 7 column with no more than 1.2).
> >
> > Q2) My former statement may be not clear enough. The operation in HL layer is hyperbolic attention, what if replacing the HL layer with an attention layer in Euclidean space?
> >
> > Q3) The hyperbolic representation and modeling have many related works in theory, and the authors provide them in the related work section. But, this paper itself does not provide any proof in theory, and the Chimera is a practical framework. Hence, the "theoretical contribution" stated in the Introduction section is not appropriate from my perspective. These contributions are more like practical contributions.

---

> > > ### Author Response · Authors · 2022-11-18
> > > **Our responses**
> > >
> > > We appreciate your response.
> > >
> > > A1）As you pointed out, based on these data sets and the quality of pre-trained model, HL did not show much improvement in these tasks. Rather, the effect of introducing HL was observed in reducing computational costs such as time to convergence and memory consumption due to dimensionality reduction.
> > >
> > > A2) Although this experimental result may not confirm a significant change, the same as above, we can say that HL requires a smaller computational cost than the Euclidean space.
> > >
> > > A3) As you point out, the practical contribution is strong, but at the theoretical contribution of Topic layer in representation learning and the Transformer framework has been supported with its practical contribution.

---

> > > > ### Comment · Reviewer_Gm2Z · 2022-11-19
> > > > **Updated review2**
> > > >
> > > > Thanks for the authors' response. As the authors pointed out, the HL does not bring a significant improvement in the current model, but it requires a smaller computational cost. It would be more suitable and convincing that improve the related statement of HL in the manuscript and add experiments to validate its efficiency.

---

### Official Review · Reviewer_hwZK · 2022-10-25

**Confidence:** 4
**Correctness:** 4
**Technical Novelty And Significance:** 2
**Empirical Novelty And Significance:** 2
**Recommendation:** 5

**Clarity, Quality, Novelty And Reproducibility:**

The presentation can be improved. Please focus on the contributions of the proposed method and present the details more clearly.

**Strength And Weaknesses:**

Strength
1. Improvements are observed.

Weaknesses
1. Only 2 datasets are used to evaluate the proposed method. More dataset, more settings and more tasks are necessary for illustration the advantages of the proposed method.

2. Fig.1 is not clear. Please use a high quality image.

3. The equations in Eq.(1) are well-known. It is not needed to mention them all.

4. About the novelty. It seems that the proposed method is just a combination of several losses.

5. How to choose lambdas in Eq(13)?

**Summary Of The Paper:**

The authors propose a framework for image-text retrieval task, by introducing matching on topic and hyperbolic space. Experiments are conducted on MSCOCO and Flickr30 datasets. Combined with some well-known models, improvements on these two datasets are shown.

**Summary Of The Review:**

See "Strength And Weaknesses".

---

> ### Author Response · Authors · 2022-11-12
> **Our current responses**
>
> Thank you for reviewing our manuscript.
> These are our responses at this stage.
> If there is anything missing, we would be very grateful and will answer them if you could let us know.
> We resubmitted the paper with increased resolution by changing the image format to PDF instead of PNG.
>
> W1) Only 2 datasets are used to evaluate the proposed method. More dataset, more settings and more tasks are necessary for illustration the advantages of the proposed method.\
> --->The selection of experimental datasets focuses on their suitability for our goal and tasks, reliability assurance of human evaluation, and reproducibility of these experiments, rather than the number of datasets.
> We need datasets in the similar domain for comparing our framework with domain-adaptive pre-training methods,
> and could not increase the number of available datasets in order to reduce the burden on evaluators, improve consistency of evaluation, and increase reproducibility for other organizations.
>
> W2) Fig.1 is not clear. Please use a high quality image.\
> --->Yes, we increased the quality of this image and resubmitted the manuscript with this image.
>
> W3) The equations in Eq.(1) are well-known. It is not needed to mention them all.\
> --->While this may be true, we believe it is easier for readers to check the correspondence with Eq.(8).
>
> W4) About the novelty. It seems that the proposed method is just a combination of several losses.\
> --->There is novelty not only in this combination,
> but also in the rationale for the combination and the task application of the components.
> In a nutshell, topics are considered as a quantized sample of the underlying feature distribution,
> and hyperbolic models allow to line up text and visual features  in a more compact alternative space.
> Firstly, the motivation is the first approach to seek an alterative space that can elucidate their semantic and complex relationships with small information loss, rather than conventional latent hidden spaces.
> As explained in 3.3,
> the topic layer is not just an application of topic models.
> We derive both several variants sharing its concept, and its training task in 3.6.
> Then we compare them in the ablation analysis, as shown in Table 4, and determine ''affine'' adapted to our goal.
> As with the hyperbolic layer,
> we select Lorentz model from common hyperbolic models and derived this layer according to Transformer and its training task in 3.6..
> Finally, we design objective function for these different spaces to learn the space in a cooperative manner,
> and align both text and image on each space.
>
> W5) How to choose lambdas in Eq(13)?\
> --->We applied Ray Tune into hyper-parameter tuning,
> and set them as $\lambda_{T},\lambda_{TLM},\lambda_{HLM},\lambda_{D}$, and $\lambda_{O}$=0.5, 0.5, 0.5, 0.5, and, 1.0, as shown under Table 3.

---

### Official Review · Reviewer_YYrR · 2022-10-25

**Confidence:** 4
**Correctness:** 3
**Technical Novelty And Significance:** 2
**Empirical Novelty And Significance:** 2
**Recommendation:** 5

**Clarity, Quality, Novelty And Reproducibility:**

The paper is novel to multi-model search,  and the experiential settings are clear to readers.

**Strength And Weaknesses:**

## Strength

- Considering hyperbolic space for multi-modal search is a novel attempt given that there is a hypernym, and an entailment relationship within words, and texts.
- The performance is also improved compared with baselines.

## Weakness

- The author only compares some small dimensional results. How about the high-dimensional performance comparisons?
- The author's comparison is limited and lacks sufficient comparison to demonstrate the superiority of his method.

**Summary Of The Paper:**

The author proposes a way to combine topic space, hyperbolic space, and Euclidean space in an end-to-end manner with learning semantic and complex relationships across modalities.

**Summary Of The Review:**

Considering hyperbolic space for multi-modal search is a novel attempt. However, there are some concerns that I mentioned before

---

> ### Author Response · Authors · 2022-11-12
> **Our current responses**
>
> Thank you for reviewing our manuscript.
> These are our responses at this stage.
> If there is anything missing, we would be very grateful and will answer them if you could let us know.
> We resubmitted the paper with increased resolution by changing the image format to PDF instead of PNG.
>
> W1) The author only compares some small dimensional results. How about the high-dimensional performance comparisons?.\
> --->Although audio data was attractive as a validation target because of its high dimensionality, only the comparison results of image data are presented in this study due to its reproducibility for other researchers and the frequency of citation.
>
> W2) The author's comparison is limited and lacks sufficient comparison to demonstrate the superiority of his method.\
> --->This may appear so because our experimental strategy is not only to follow the design of previous studies, but also to present results narrowly compared to SOTA (i.e., VD-BERT and ViT) in order to improve reproducibility for other researchers and to reduce the burden on reviewers.

---

### Decision · Program_Chairs · 2023-01-20

**Decision:**

Reject

**Justification For Why Not Higher Score:**

N/A

**Justification For Why Not Lower Score:**

N/A

**Metareview: Summary, Strengths And Weaknesses:**

This paper proposed a method to combine topic space, hyperbolic space, and Euclidean space in an end-to-end manner with learning semantic and complex relationships across modalities. While the paper contains some interesting idea, reviewers raised major weakness concerns about weak empirical evaluations on limited small datasets, and lack of comparisons with more recent state-of-the-art methods, missing ablation study, etc. While the authors tried to respond to the review questions, some of major review concerns remain and the overall quality of this work is below the acceptance bar.